# *De novo* assembly of the freshwater prawn *Macrobrachium carcinus* brain transcriptome for identification of potential targets for antibody development

Jonathan L. Crooke-Rosado[1,2], Sara C. Diaz-Mendez[3⊙], Yamil E. Claudio-Roman[3⊙], Nilsa M. Rivera[1,2], Maria A. Sosa[1,2]*

1 Department of Anatomy & Neurobiology, School of Medicine, Medical Sciences Campus, University of Puerto Rico, San Juan, Puerto Rico, 2 Institute of Neurobiology, Medical Sciences Campus, University of Puerto Rico, San Juan, Puerto Rico, 3 Department of Biology, Cayey Campus, University of Puerto Rico, Cayey, Puerto Rico

⊙ These authors contributed equally to this work.
* maria.sosa@upr.edu

**Data Availability Statement:** All relevant data are within the paper and its Supporting information

## Abstract

Crustaceans are major constituents of aquatic ecosystems and, as such, changes in their behavior and the structure and function of their bodies can serve as indicators of alterations in their immediate environment, such as those associated with climate change and anthropogenic contamination. We have used bioinformatics and a *de novo* transcriptome assembly approach to identify potential targets for developing specific antibodies to serve as nervous system function markers for freshwater prawns of the *Macrobrachium* spp. Total RNA was extracted from brain ganglia of *Macrobrachium carcinus* freshwater prawns and Illumina Next Generation Sequencing was performed using an Eel Pond mRNA Seq Protocol to construct a *de novo* transcriptome. Sequencing yielded 97,202,662 sequences: 47,630,546 paired and 1,941,570 singletons. Assembly with Trinity resulted in 197,898 assembled contigs from which 30,576 were annotated: 9,600 by orthology, 17,197 by homology, and 3,779 by transcript families. We looked for glutamate receptors contigs, due to their main role in crustacean excitatory neurotransmission, and found 138 contigs related to ionotropic receptors, 32 related to metabotropic receptors, and 18 to unidentified receptors. After performing multiple sequence alignments within different biological organisms and antigenicity analysis, we were able to develop antibodies for prawn AMPA ionotropic glutamate receptor 1, metabotropic glutamate receptor 1 and 4, and ionotropic NMDA glutamate receptor subunit 2B, with the expectation that the availability of these antibodies will help broaden knowledge regarding the underlying structural and functional mechanisms involved in prawn behavioral responses to environmental impacts. The *Macrobrachium carcinus* brain transcriptome can be an important tool for examining changes in many other nervous system molecules as a function of developmental stages, or in response to particular conditions or treatments.

files. Data are also available in a public repository at http://www.ncbi.nlm.nih.gov/bioproject/716066.

**Funding:** This work was supported by the National Science Foundation (NSF) Centers of Research Excellence in Science and Technology (CREST) awards HRD-1137725 and HRD-1736019 (https://www.nsf.gov/funding/pgm_summ.jsp?pims_id=6668; project support to MS), the National Institutes of Health (NIH) – National Institute on Minority Health and Health Disparities (NIMHD) Research Centers in Minority Institutions (RCMI) program (award MD007600 – https://www.nimhd.nih.gov/programs/extramural/research-centers/rcmi/; support of core facilities at Institute of Neurobiology), and the National Institutes of Health (NIH) – National Institute of General Medical Sciences (NIGMS) Research Training Initiative for Student Enhancement (RISE) Program (R25 GM-061838 – https://www.nigms.nih.gov/training/RISE; fellowship support to JC). The funders had no role in study design, data collection and analysis, decision to publish, or preparation of the manuscript.

**Competing interests:** The authors have declared that no competing interests exist.

## Introduction

Crustaceans are major constituents of aquatic ecosystems, living in different habitats depending on their needs, and playing an ecological role in their habitat as shredders and nutrient cyclers [1]. They are beneficial to the ecosystem as they participate in consuming decaying vegetables and animal bodies in the water [2]. Their frequent use as bio-indicators and bio-monitors [3] in various aquatic systems to assess water and ecosystem quality is one of their main contributions to ecological studies. Indicators of environmental stress in water include relative abundance and diversity, feeding activity, drifting, changes in metabolism, molting and growth, immune functions, reproductive capacity and locomotion [4]. Since the beginning of 2010, RNA sequence analysis and transcriptomic approaches have impacted and redefined crustacean research. Characterization of mechanisms and molecules associated with behavioral and metabolic or physiological changes is a current research focus given the commercial and environmental relevance of these invertebrate animals.

Gene expression from gills [5–8], hepatopancreas [5, 9] and muscles [7, 10] is commonly used for transcriptomic analysis of growth, metabolism, immunity and changes due to environmental stressors in many crustaceans. For instance, Jung and colleagues [6] used *Macrobrachium rosenbergii* transcriptomes of these tissues to characterize single nucleotide polymorphisms (SNP) in 23 growth-related candidate genes to facilitate the improvement of growth performance in cultured animals. In this same species, regulation in the metabolism of essential molecules such as amino acids, carbohydrates, lipids, vitamins and co-factors, glycans, terpenoids and polyketides, was demonstrated by pathway enrichment using available transcriptomes [7]. In another study, innate immunity against White Spot Syndrome Virus (WSSV) was assessed evaluating the presence of Single Nucleotide Polymorphisms (SNPs) in the shrimp *Litopenaeus vannamei* [11]. The importance of crustaceans and their environmental interactions and effects has also been emphasized in several studies. Eighteen differentially expressed genes related to responses to stimuli, transferase activity, oxidative phosphorylation, as well as adverse effects at a structural level in the hepatopancreas, gills and muscle tissues were identified in *Macrobrachium nipponense*, following seven days of exposure to chronic hypoxia as an environmental stressor [12]. Moreover, more than 36 genes in the mud crab *Scylla paramamosain* were shown to have changed expression profiles after exposure to the heavy metal cadmium [13]. However, the list of transcriptomic studies narrows down when looking at structural and functional properties of the nervous system of crustaceans.

Crustacean species used as research subjects in neural transcriptome studies include the water flea *Daphnia pulex* [14], gazami crab *Portunus trituberculatus* [15], American lobster *Homarus americanus* [16–18], red swamp crayfish *Procambarus clarkii* [19, 20], eastern rock lobster *Sagmariasus verreauxi* [21], copepod ectoparasite *Caligus rogercresseyi* [22, 23] and the Macrobrachium species [24, 25]. Several studies have focused on the eyestalk, since it is a major site for the regulation of molting, reproduction, epidermal color patterns and osmosis by the production of neurohormones such as the crustacean hyperglycemic hormone (CHH), crustacean cardioactive peptide (CCAP), eclosion hormone, and pigment-dispersing hormone (PDH), among others. Other than the work by Núñez-Acuña and colleagues [22], who evaluated changes in the neurotransmission system of the copepod ectoparasite *Caligus rogercresseyi* induced by the xenobiotic drug deltamethrin (DM) and by azamethiphos (AZA) in pesticides, studies that outline nervous system impairments using a transcriptomics approach are limited. Núñez-Acuña´s group [22] showed that the glutamatergic synaptic pathway of the parasite was affected by both DM and AZA, causing a down-regulation of the glutamate-ammonia ligase, and that DM activates a metabotropic glutamate receptor that is a suggested inhibitor of neurotransmission.

Glutamate is very important in crustacean neurotransmission, being generally involved in excitatory influences in both the central and peripheral nervous systems. To date there is no specific antibody against crustacean glutamate receptors, making it difficult to fully elucidate structural and functional properties of glutamate synapses. However, a few studies have reported ionotropic NMDA-like glutamate receptor immunoreactivity in crustacean models using antibodies raised against mammalian NMDA receptors [26–28]. Feinstein and colleagues [26] showed NMDA antibody staining at the presynaptic membranes of neuromuscular junctions in the crayfish *Procambarus clarkii*. In another study, Gallus and colleagues [27] showed NMDA-like immunoreactivity in the peripheral nervous system and non-neuronal structures of the cyprid larval stage of *Balanus amphitrite*, specifically in the thoracic appendages, suggesting an NMDA role in neuromuscular control. Moreover, Hepp and colleagues [28] also identified NMDA-like immunoreactivity in most of the central nervous system ganglia (eyestalk, brain and thorax) of the crab *Neohelice granulata*, establishing a correlation with previously described memory processes in this animal model. Unfortunately, in our hands some of the commercially available glutamate receptor antibodies used with nervous system tissue of *Macrobrachium* prawns appear to show non-specific binding.

In the present study, we use bioinformatics and a *de novo* transcriptome assembly approach to identify potential targets for developing antibodies specific to glutamate receptors in freshwater prawns of the *Macrobrachium* spp to serve as nervous system markers. We extracted total RNA from brain ganglia of the freshwater prawn *Macrobrachium carcinus* and identified sequences related to glutamate receptors, looked for conserved sequences between different biological organisms, and designed antibodies according to antigenicity analysis of sequences. The long-term goal of this study is to develop nervous system markers that work well in *Macrobrachium* spp that can be used to monitor changes in structural and functional neural properties as a result of environmental impacts.

## Methodology

### Animal husbandry

For transcriptome assembly, four adult male *Macrobrachium carcinus (M. carcinus)* freshwater prawns were caught in non-urban rivers in Yabucoa, Puerto Rico, as authorized through permit 2016-IC-145 (R-VS-PVS15-SJ-00560–08092016) of the Department of Environmental and Natural Resources of the Government of Puerto Rico. Animals were maintained in the lab in 5-gallon tanks with continuously filtered and aerated water under a 12:12-h light/ dark cycle for ten days of habituation prior the start of the experiment. High protein (>40%) pelleted Purina chow was administered once every two days. Water temperature was maintained at 26–28 ˚C and the pH adjusted to 7.2–7.5. All procedures were carried out in strict accordance with the recommendations in the Guide for the Care and Use of Laboratory Animals of the National Institutes of Health. The protocol was previously approved by the University of Puerto Rico Medical Sciences Campus´ Institutional Animal Care and Use Committee (IACUC #A3240113).

### RNA isolation, sequencing and data processing

Total RNA was extracted from brain (supraesophageal) ganglia of the ventral nerve cord using RNAqueous™ Total RNA Isolation Kit (Invitrogen, Cat. # AM1912), following the manufacturer's instructions. Briefly, brain ganglia were dissected, and RNases inactivated by placing the tissue in RNA *later* Solution. The samples were homogenized in Lysis/Binding Solution (10–12 uL/mg), and an equal amount of 64% ethanol was added and mixed. The lysate/ethanol mixture was applied to a filter cartridge and centrifuged at 14,000 rpm. The filtered mixture

was washed with 700 uL of Wash Solution #1, followed by washes with 2 x 500 uL of Wash Solution #2/3. RNA was eluted with 40–60 μL preheated Elution Solution, following another elution with a second 10–60 μL aliquot of Elution Solution. An *Illumina Next generation Sequencing HiSeq2000* was employed for sequence analysis by the company MacroGen. The HiSeq2000 utilized HiSeq Control Software (HCS) v2.2.38 to generate raw images for system control and base calling through Real Time Analysis (RTA) v1.18.61.0, a software for integrated primary analysis. The base calls (BCL) binary was converted into FASTQ utilizing illumina package bcl2fastq (v1.8.4).

## Transcriptome assembly

De novo assembly of the *M. carcinus* transcriptome was performed from the raw RNA-Seq data with The Eel Pond mRNAseq Protocol (khmer-protocols 0.8.4). Quality of the sequences was assessed with FASTQC and the trimming of Illumina TruSeq3-PE.fa adapters from sequences was performed with the *Trimmomatic* sequence analysis tool. Trinity software v2.0.4 was used for de novo transcriptome assembly. Trinity consists of three parts: 1) Inchworm, to assemble initial contigs; 2) Chrysalis, to build the de Bruijn graphs; and 3) Butterfly, to resolve alternative splicing independently for each cluster. Subsequently, transcripts were preliminarily annotated through the National Center for Biotechnology Information (NCBI) Basic Local Alignment Search Tool (BLAST), against a UniProtKB's/Swiss-Prot database using a mouse RefSeq (reference sequence). BLAST best hit and reciprocal best hit was used to assign names to sequences by calculation of putative homology and orthology, respectively.

## Transcriptome analysis

A direct search for transcripts related to glutamate receptors within the newly derived transcriptome was performed. Multiple Sequence Alignments between the annotated transcripts for glutamate receptors and known sequences for the same glutamate receptor in other biological organisms found through NCBI BLAST was performed with the "EMBL-EBI MUSCLE Multiple Sequence Alignment" web server. This online tool identifies conserved sequences of proteins or DNA/RNA, and assigns colors or consensus symbols to the identified conserved regions, according to their similarity percentage. A blank space ("") indicates a poorly conserved region or one with no similarity. A period (".") indicates a somewhat similar region. A colon (":") indicates a very similar region, and an asterisk ("*") indicates good conservation (same amino acid or nucleotide).

## Antigenicity assessment of transcript sequences

Amino acid sequences from transcripts with conserved similarity regions between species were submitted to an online program from the *Immunomedicine Group* of "Universidad Complutense de Madrid" to predict, using the Kolaskar and Tongaonkar [29] method, segments with antigenic properties to elicit an antibody response. Predictions are based on the occurrence of amino acid residues in experimentally known segmental epitopes. Segments are reported if they have a minimum size of 8 residues. These predictions were sent to GL Biochem (Shanghai, China) for further analysis and antibody production.

## Results

### RNA sequencing, *de novo* assembly and annotation

To obtain potential neural markers in the prawn CNS and neuromuscular junction, a cDNA library was constructed from purified total RNA from brain ganglia of adult *M. carcinus*

**Table 1. Statistics of *M. carcinus* RNA sequencing, transcriptome assembly and annotation.**

| Sequencing, Transcriptome Assembly and Annotation Outputs | |
|---|---|
| Raw reads | 103,603,508 |
| Clean reads | 97,202,662 |
| % Q30 | 92.01 |
| % GC | 42.22 |
| Number of singletons | 1,941,570 |
| Assembler | Trinity |
| Number of contigs | 197,899 |
| $N_{50}$ (bp) | 1,911 |
| Mean contig length (bp) | 875.44 |
| Median contig length (bp) | 381 |
| Longest contig (bp) | 28,864 |
| Shortest contig (bp) | 201 |
| Greater than 2K (bp) | 21,303 |
| Total assembled bases | 173,249,903 |
| Total contigs annotated | 30,576 |

freshwater prawns. Further sequencing through the NGS Illumina HiSeq 2000 platform yielded 103,603,508 raw reads, for a total of 10,463,954,308 bases. The percentage of guanine to cytosine ratio (%GC) was 44.22% and the phred-score-above-30 (Q30) was 92.01%, the latter indicating overall high read quality, since the Q30 score is indicative of a 99.9% of base call accuracy, or 0.01% probability of error chances. FASTQC quality control and trimming of Illumina TruSeq3-PE.fa adapters from raw reads resulted in a total of 97,202,662 clean reads, distributed as 47,630,546 paired right and left, and 1,941,570 singletons, which were used for *de novo* transcriptome assembly. Statistics for the RNA sequencing and others important for the assembly are shown in Table 1.

*De novo* assembly was performed with Trinity assembler, resulting in 197,899 contigs representing 173,249,903 total assembled bases, with a mean contig length of 875 base pairs. N50 contig length was 1,911. This is traditionally defined as the shortest sequence length, such that half of the total sequence output length is included in sequences that are shorter [30]. The longest contig and the shortest contig were of 28,864 and 201 base pairs, respectively, with approximately 15% of the contigs greater than 2,000 base pairs in length (Table 1). Putative homology and orthology of the contigs were assigned through BLASTs best hits in both directions (blastx and tblastn), and reciprocal best hit analysis, respectively, using a UniProtKB's/Swiss-Prot database with a mouse RefSeq. This resulted in a total of 30,576 contigs successfully annotated (e value $< 10^{-4}$; Fig 1). From those, 17,197 contigs (56.2%) were annotated by putative homology, 9,600 (31.3%) were annotated by orthology, and 3,779 were annotated by transcript families. The *de novo* annotated transcriptome is available at http://www.ncbi.nlm.nih.gov/bioproject/716066 or S1 File.

## Surveying for glutamate receptors

A direct search within the new transcriptome using "glutamate receptor" as keywords generated a total of 183 contigs (overlapping DNA sequences used to make a physical map that reconstructs the original DNA sequence of a chromosome or a region of a chromosome; National Human Genome Research Institute) associated with glutamate receptor transcripts: 133 ionotropic, 32 metabotropic, and 18 not specified as one of the previous categories (Fig 2).

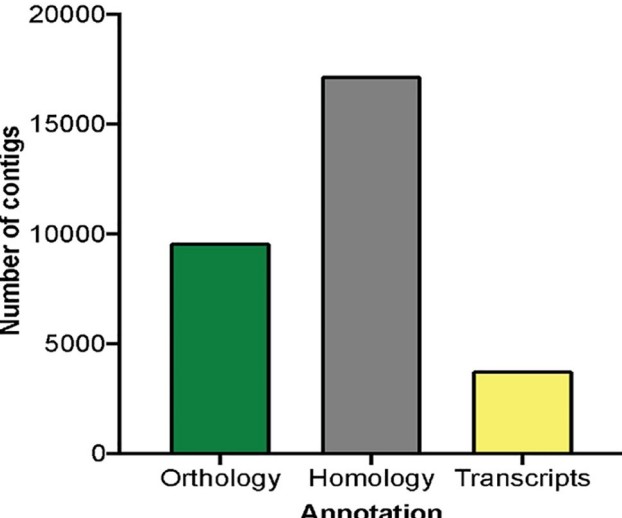

**Fig 1. De novo transcriptome annotation homology.** A total of 30,576 contigs was annotated by blastx and tblastn. Reciprocal best hits yield 9,600 (31.3%) contigs annotated by orthology; green bar. BLAST best hits yield 17,197 (56.2%) contigs annotated by putative homology; gray bar. The remaining 3,779 (12.4%) contigs were annotated by transcript families; yellow bar.

Those contigs annotated as ionotropic glutamate receptors (GluR) include: GluR 1–4, GluR delta 1 and 2; GluR kainate 1–5; NMDA 1, 2A, 2B, 3B, 2D. Those contigs related to metabotropic glutamate receptors include: mGluR; Grm2–6 and Grm8. Information about homologous species and nucleotide sequence of these contigs are available in S2 File.

## Multiple sequence alignments and antigenicity analysis

Through multiple sequence alignments using the EMBL-EBI "MUSCLE" Multiple Sequence Alignments web server, we looked for conserved regions within the contigs sequence, suggesting likely important functional domains, as compared to the same protein in other biological species selected in the NCBI Protein database. From the list in this database, we selected three

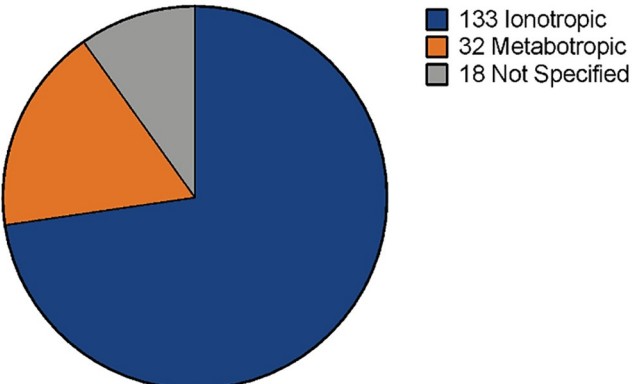

**Fig 2. Contigs annotated as glutamate receptors in M. carcinus de novo transcriptome.** Searching for glutamate receptor sequences within the new transcriptome resulted in 133 contigs identified as ionotropic glutamate receptors, 32 contigs identified as metabotropic glutamate receptors and 18 contigs identified as glutamate receptors without further classification.

to four different sequences that represent the same protein fully annotated and in which at least one speciation phenomena had occurred. For this step, nucleotide sequences of contigs were converted to one symbol amino acid sequences using the "ExPASy Translate Tool" online. This analysis helped us prioritize which contigs have high potential for further antibody design.

Five contigs generated reliable multiple sequence alignments when compared to other biological species. Fig 3 shows the first contig, identified as id = 128409 and corresponding to transcript tr = 427041, with nucleotide length of 382 bp, and annotated as mGluR in *Drosophila melanogaster* (Fig 3a). This contig was compared to mGluR in the branchiopod *Daphnia pulex* (EFX87083.1), the insect *Linepithema humile* (XP_012230284.1) and the fruit fly *D. melanogaster* (NP_524639.2), as shown in Fig 3b. Alignments show approximately 60% of the contig sequence conservation, denoted by asterisks "*" in the bottom consensus line. Antigenicity analysis achieved using the "Immunomedicine Group" web server resulted in 6 predicted regions within the contig sequence with antigenic properties, averaging an antigenic propensity of 1.0790 (Fig 3c). A whole sequence having an average antigenic propensity above 1.0 means that amino acids (residues) within the sequence, when examined individually and also showing antigenic propensity above 1, are predicted to elicit antigenic properties. According to this analysis, a good peptide sequence for recognition with an antibody against this contig is: AVKTRKIPENFNESK, located at the cytoplasmic side, between residues 68–82, as highlighted in red in Fig 3a and enclosed by the red rectangle in Fig 3b. A 3D model reconstruction of the contig sequence´s tertiary structure was made with Protein Homology/AnalogY Recognition Engine (Phyre2) web services for location and physical visualization of the proposed antigenic peptide sequence, shown in red in Fig 3d.

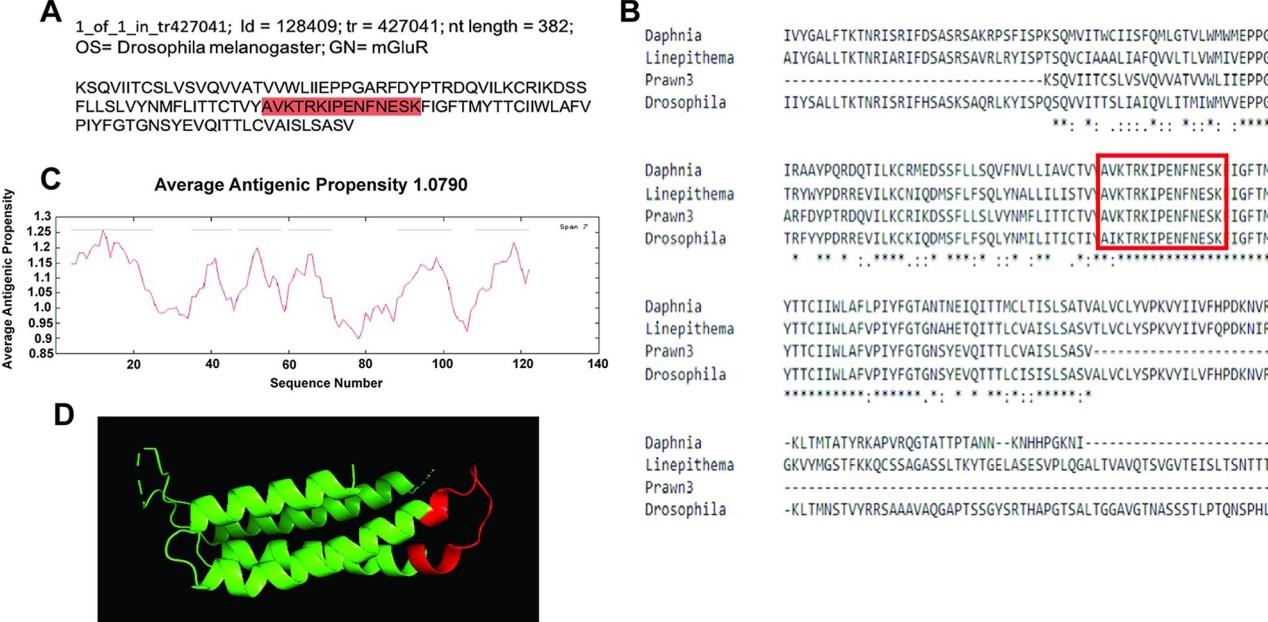

**Fig 3. Contig annotated as metabotropic glutamate receptor 1 (mGluR1) analysis for antibody production.** A) Contig identifiers for mGluR1 in the *M. carcinus* transcriptome and its sequence translated into amino acids for further analysis. B) Multiple sequence alignments of the prawn´s contig were compared to the same protein in other species: branchiopod *D. pulex*, insect *L. humile*, and the fruit fly *D. melanogaster*. Regions of conserved amino acids are shown within the red box. C) Average antigenic propensity value was 1.0790, showing 6 determinants (peaks in graph) within the sequence that are good candidates for antigenic response. D) A 3D reconstruction model showing tertiary structures of the prawn mGluR1 contig sequence. The red highlight in A, rectangle in B, and segment in D represent the selected sequence for antibody production.

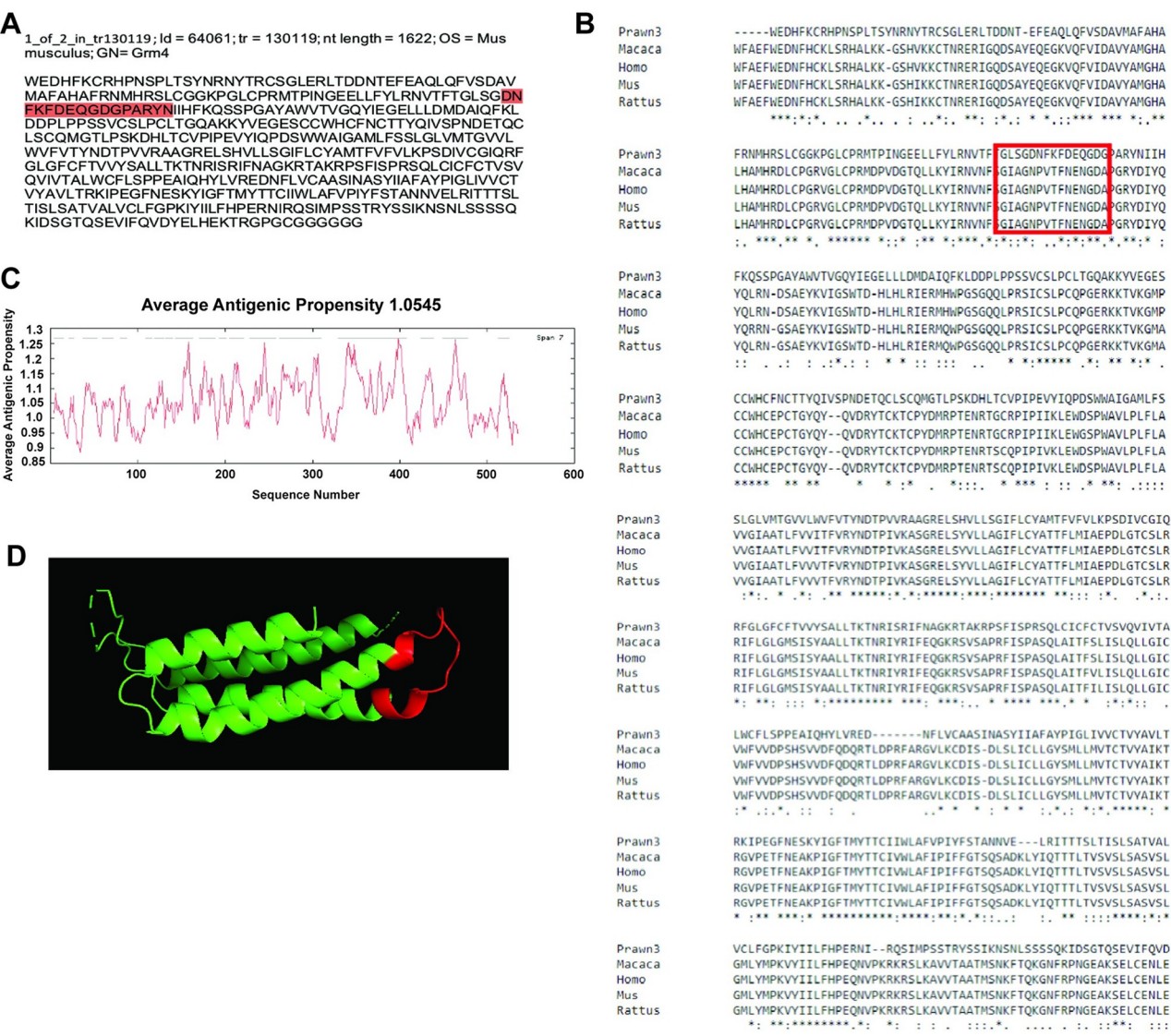

**Fig 4. Contig annotated as metabotropic glutamate receptor 4 (mGluR4) analysis for antibody production.** A) Contig identifiers for mGluR4 in the *M. carcinus* transcriptome and its sequence translated into amino acids for further analysis. B) Multiple sequence alignments of the prawn´s contig compared to the same protein in other species: non-human primate *M. mulatta*, human primate *H. sapiens*, rodent *M. musculus*, and rodent *R. norviscus*. Regions of conserved amino acids are shown within the red box. C) Average antigenic propensity value was 1.0545, showing 18 determinants (peaks in graph) within the sequence that are good candidates for antigenic response. D) A 3D reconstruction model showing tertiary structures of the prawn mGluR4 contig sequence. The red highlight in A, rectangle in B, and segment in D represent the selected sequence for antibody production.

The second contig that shows reliable sequence conservation is the one identified as id = 64061 and corresponding to transcript tr = 130119, with nucleotide length of 1622 bp, and annotated as metabotropic glutamate receptor 4 (mGluR4) in the rodent *Mus musculus* (Fig 4a). This contig sequence was compared to mGluR4 in the rodent *M. musculus* (EDL22539.1), another rodent *Rattus norvegicus* (NP_073157.1), the primate *Macaca mulatta* (XP_014991606.1) and the human primate *Homo sapiens* (ABY87923.1) as shown in Fig 4b. Nearly 40% of the contig sequence shows conservation, denoted by asterisks in the consensus line. Average antigenic propensity of the entire sequence was 1.0545, resulting in a total of 18 predicted regions elucidating antigenic properties (Fig 4c). After antigenicity analysis, the

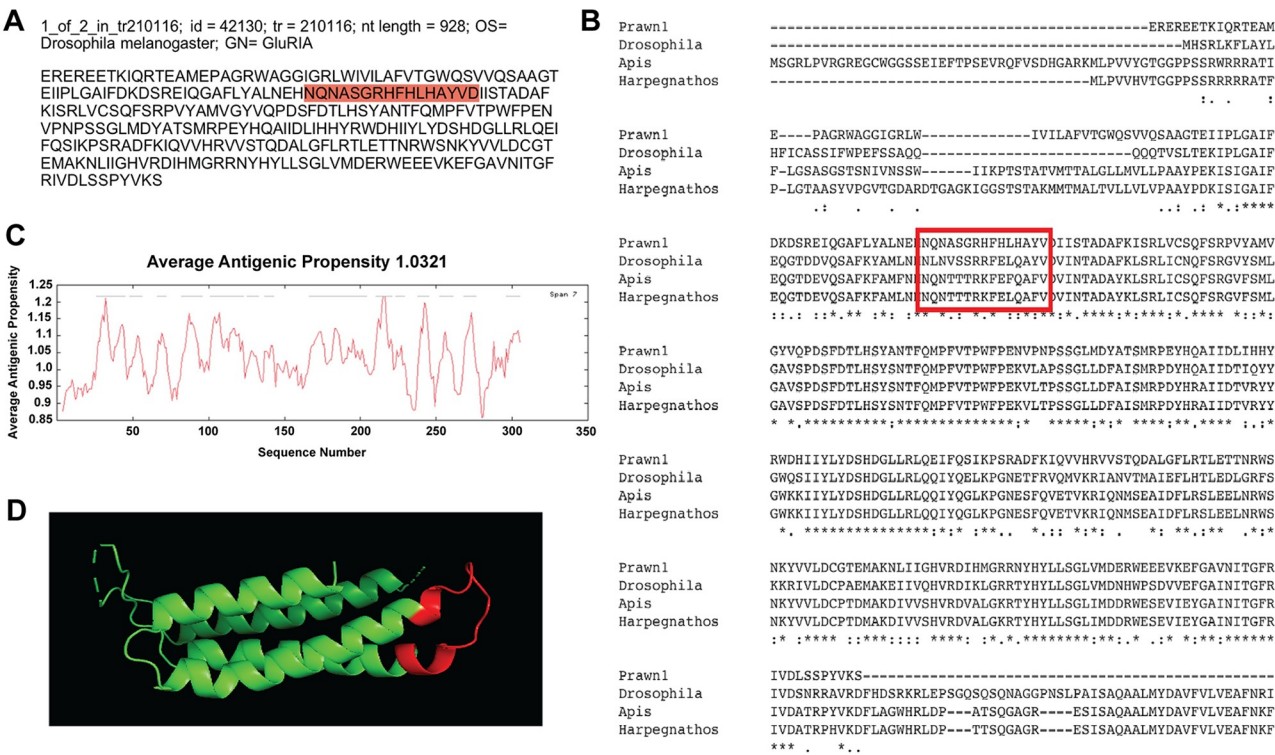

**Fig 5. Contig annotated as ionotropic glutamate receptor A-1 (GluRIA-1) analysis for antibody production.** A) Contig identifiers for GluRIA-1 in the *M. carcinus* transcriptome and its sequence translated into amino acids for further analysis. B) Multiple sequence alignments of the prawn´s contig were compared to the same protein in other species: the fruit fly *D. melanogaster*, European bee *A. mellifera*, and the Indian jumping ant *H. saltator*. Regions of conserved amino acids are shown within the red box. C) Average antigenic propensity value was 1.0321, showing 14 determinants (peaks in graph) within the sequence that are good candidates for antigenic response. D) A 3D reconstruction model showing tertiary structures of the prawn GluRIA-1 contig sequence. The red highlight in A, rectangle in B and segment in D represent the selected sequence for antibody production.

sequence selected for recognition with an antibody against this contig was: DNFKFDEQGDG PARYN, located at the extracellular side between residues 95–115 highlighted in red in Fig 4a and enclosed by the red rectangle in Fig 4b. A 3D model reconstruction of the contig sequence's tertiary structure for location and physical visualization of the proposed antigenic peptide sequence is shown in red in Fig 4d.

The third contig that showed good sequence conservation is id = 42130, one of two variants of transcript tr = 210116, with nucleotide length of 928 bp annotated as ionotropic glutamate receptor 1 (GluRIA) in the fruit fly *D. melanogaster* (Fig 5a). This contig sequence was compared to GluRIA in *D. melanogaster* (AAF50652.2), the European bee *Apis mellifera* (XP_006565171.1) and the Indian jumping ant *Harpegnathos saltator* (XP_011137572.1), as shown in Fig 5b. Almost 50% of the contig sequence shows conservation, denoted by asterisks in the consensus line. Average antigenic propensity of the entire sequence was 1.0321, resulting in a total of 14 predicted regions elucidating antigenic properties (Fig 5c). The sequence selected for recognition with an antibody against this contig was: NQNASGRHFHLHAYVD, located at the extracellular side between residues 77–92 (Fig 5a and 5b; red highlight and red rectangle respectively). None of the peptide sequences suggested by GL Biochem were located in the most conserved region of the contig. However, the selected peptide sequence has very similar amino acid content compared to the other sequences and a good antigenic index. A 3D model reconstruction of the contig sequence's tertiary structure for location and physical visualization of the proposed antigenic peptide sequence is shown in red in Fig 5d.

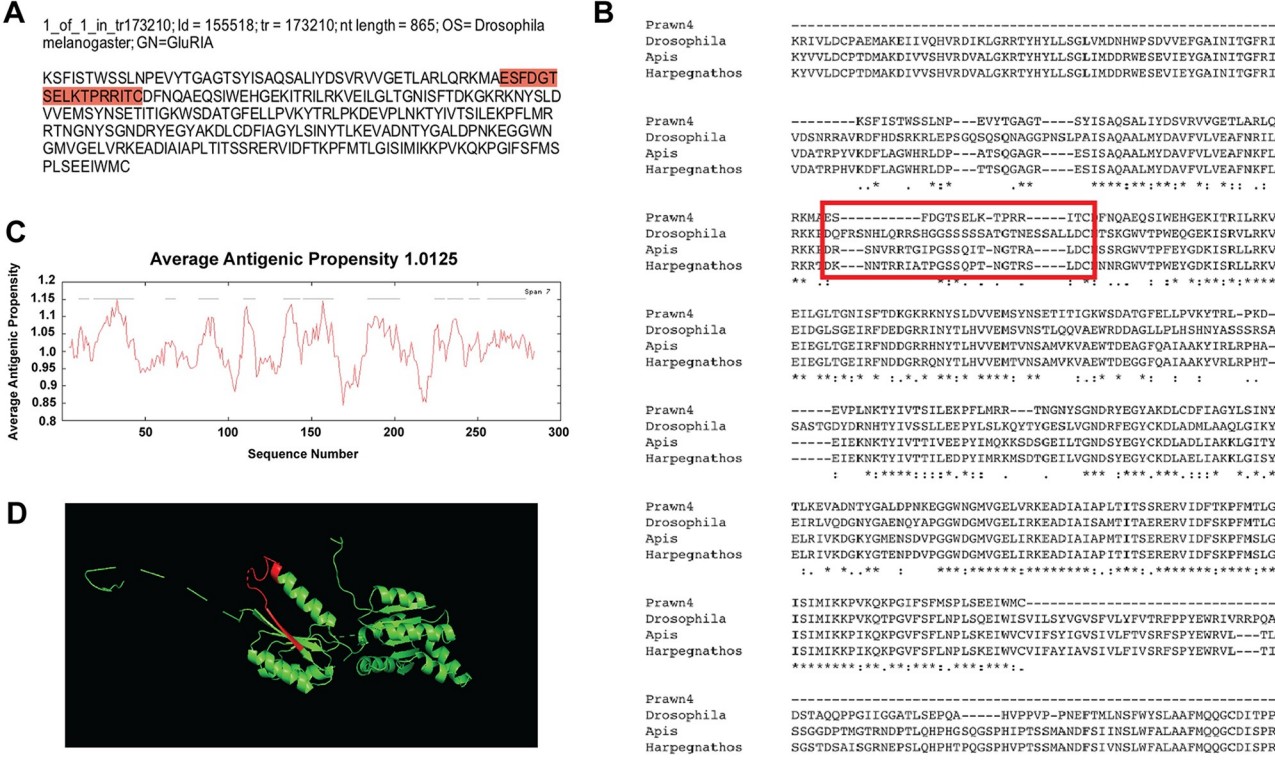

**Fig 6. Contig annotated as ionotropic glutamate receptor A-2 (GluRIA-2) analysis for antibody production.** A) Contig identifiers for GluRIA-2 in the *M. carcinus* transcriptome and its sequence translated into amino acids for further analysis. B) Multiple sequence alignments of the prawn´s contig compared to the same protein in other species: the fruit fly *D. melanogaster*, European bee *A. mellifera*, and the Indian jumping ant *H. saltator*. Regions of conserved amino acids are shown within the red box. C) Average antigenic propensity value was 1.0125, showing 12 determinants (peaks in graph) within the sequence that are good candidates for antigenic response. D) A 3D reconstruction model showing tertiary structures of the prawn GluRIA-2 contig sequence. The red highlight in A, rectangle in B and segment in D represent the selected sequence for antibody production.

The fourth contig with regional conservation is id = 155518, a single variant of tr = 173210, with nucleotide length of 865 bp and annotated also as ionotropic glutamate receptor 1 (GluRIA) in *D. melanogaster* (Fig 6a). Results of multiple sequence alignments against the same species as the third contig (id = 42130) are shown in Fig 6b. This time, approximately 45% of the sequence shows conservation, denoted by asterisks in the consensus lines. Average antigenic propensity of the contig sequence was 1.0125, resulting in a total of 12 predicted regions elucidating antigenic properties (Fig 6c). The peptide sequence selected for antibody recognition was: ESFDGTSELKTPRRITC, located at the extracellular side between residues 50–66 (Fig 6a and 6b; red highlight and red rectangle respectively). Although this sequence did not turn out to be in a conserved region, its selection was based on the antigenicity analysis conducted by the company GL Biochem. The 3D model reconstruction of the contig sequence's tertiary structure for location and physical visualization of the proposed antigenic peptide sequence is shown in red in Fig 6d.

Our last contig showing regions with amino acid conservation is id = 37464, corresponding to the transcript tr = 866271 with nucleotide length of 2910 bp and annotated against ionotropic glutamate receptor NMDA 2B in the dog *Canis familiaris* (Fig 7a). This contig sequence was compared to NMDA 2B in *M. musculus* (XP_017176885.1), *H. sapiens* (NP_000825.2), and *M. mullata* (XP_001088140.1), as shown in Fig 7b. In this case, 51% of the contig sequence showed good conservation having asterisks at the bottom consensus line. Average antigenic propensity of the entire sequence was 1.0321, resulting in a total of 37 predicted regions

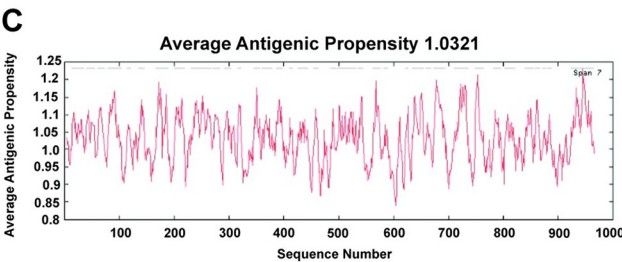

A)
1_of_1_in_tr866271; id=37464; tr=866271; nt length = 2910; OS= Canis familiaris; GN= GRIN2B

LPSSPSSATFRRRSVRLLTPQPPPQAMLLSRHSHLSRTKDVSLHSSSPSSSCSSSS
SSNSSPSISLLHSPSTPAAQWRRRLAVAIFTSLLLLQFPQDCNSYLNTYGRKEEEHT
ITIGIIHPKTTFRQRLYTKAINDAVGALLKLDLKFMKTSFIANSTTPMDINPSPTSILEC
LCKRFLPENVSAIIFLTRTETYGRNTASAQYFLQLAGYLGIPVIAWNADNSGLEQAA
QSGLRIQLAPSVHHQAAAMLSILVRYQWHSLSIVTSQIAGHTDFVQAVRDQVAQHK
ENHMGKFVIIDTIIVDDIRTSLLRLKNSEARIILLYSTRDEATDIMKEANGLGLTDKNYI
WIVTQSVVGDREVPSNELPVGMLGVHFDTSLDSLTSNIPTAIEVFAYAVEKFVNQT
RFTPSDLNPRLSCDDHRNAKWGLGYTFYRHLRNVSIPRHSSSISFNTDGTRKDVEL
KIVNLREGTSNNRVWEEIGVWRSWEGEGLSVNDIVWPGNGHVPPQGVPEKFHLKI
TYLEEPPYVNLAPADPVTNQCSANRGVLCKIPRKKPGEAGAGNTSTSASTTMCCS
GFCIDLLEKFANDLGFSYELMRVADGKWGTFDQTNNRWNGLIGELLQREGEGAE
MVLTSLKINKERESVVDFTVPFLESGIAIVVAKRTGIISPTAFLEPFDTASWMLVAFV
AIQVAALTIFLFEWLSPGGYNMRMAPPRDHKFSLFRTYWLVWAVLFQAAVHVDCP
RGFTARFMANMWAMFAVVFLAIYTANLAAFMITREEFHDFSGINDTRLQYPSTVSP
AFKFGTVDETNSAMVLKRHHPLMFSYMVRHKYNRDSVQDGIKSIKKNELDAFIYDA
TVLDYRVGQDDDCQILTVGSWYSMSGYGLAFPRGSKYLSLFNNKLMSYKDNGDIE
RLQRFWMTG TCKPKKQQRRASEP LAIEQFLSAYLLLGIGMVIAIVLLALEHIYFKYVR
KHLAKKDSG

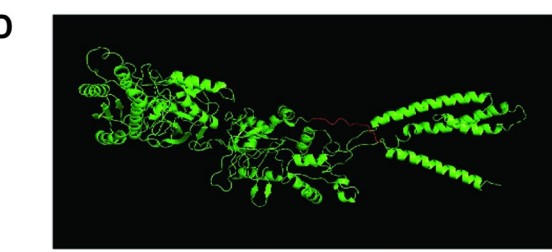

B)

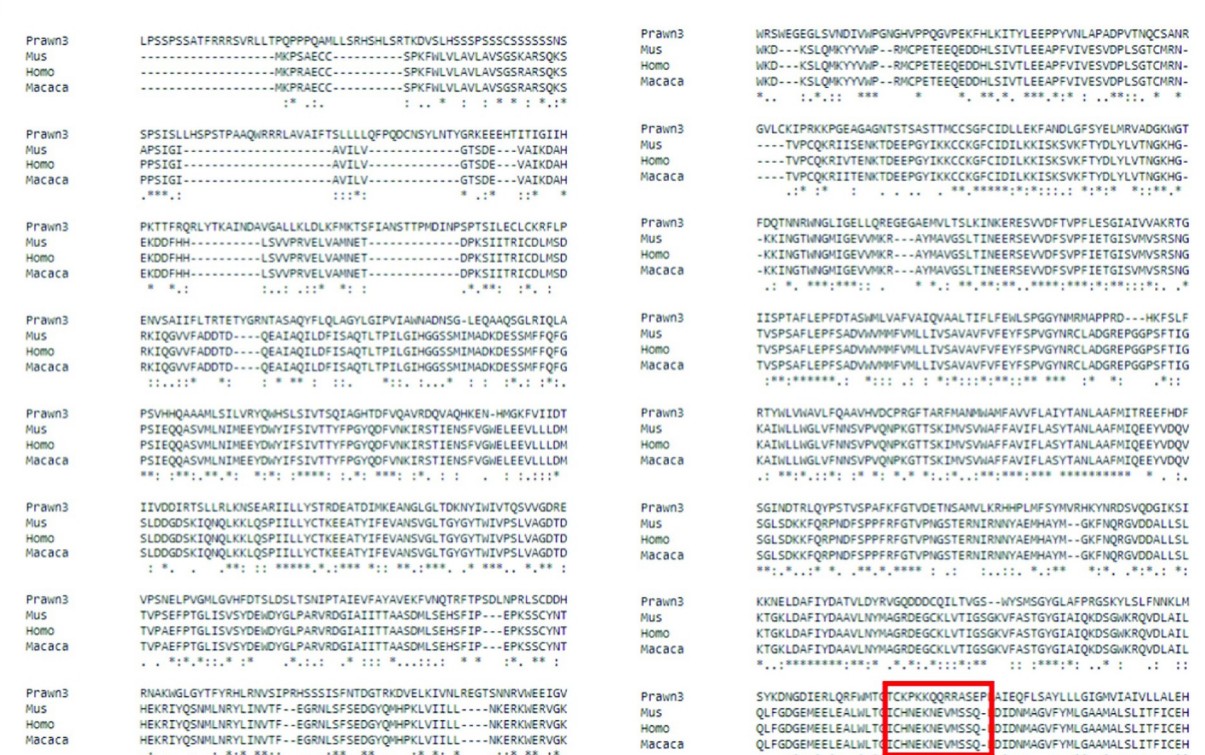

**Fig 7. Contig annotated as ionotropic glutamate receptor NMDA 2B analysis for antibody production.** A) Contig identifiers for NMDA 2B in the *M. carcinus* transcriptome and its sequence translated into amino acids for further analysis. B) Multiple sequence alignments of the prawn´s contig compared to the same protein in other species: a rodent *M. musculus*, human primate *H. sapiens*, and the non-human primate *M. mulatta*. Regions of conserved amino acids are shown within the red box. C) Average antigenic propensity value was 1.0321, showing 37 determinants (peaks in graph) within the sequence that are good candidates for antigenic response. D) A 3D reconstruction model showing tertiary structures of the prawn NMDA 2B contig sequence. The red highlight in A, rectangle in B and segment in D represent the selected sequence for antibody production.

elucidating antigenic properties, in part due to the great length of the sequence (Fig 7c). After antigenicity analysis, the sequence selected for recognition with an antibody against this contig was: TCKPKKQQRRASEP, located at the extracellular side between residues 912–925 (Fig 7a and 7b; red highlight and red rectangle respectively). The 3D model reconstruction of the

contig sequence tertiary structure for location and physical visualization of the proposed antigenic peptide sequence is shown in red in Fig 7d.

Finally, the selected antigenic sequences were sent to GL Biochem to proceed with the production of rabbit raised polyclonal antibodies which are presently being tested as markers in the prawn´s nervous system.

## Discussion

From the large number of species that are part of the genus *Macrobrachium*, only four have been previously used to develop genomic databases: *M. rosenbergii* [10], *M. nipponense* [31], *M. olfersii* [32], and *M. australiense* [33]. To our knowledge, this is the first study that reports genomic information for the species *M. carcinus*. This work demonstrates a high quality *de novo* transcriptome assembly using brain tissue from this crustacean. The *M. carcinus* prawn is important due to its commercial value through the practice of fishing, being a primary species in Neotropical rivers [34], including the tropical regions of North and Central America, and all of South America´s temperate zones. This transcriptome dataset, although limited to brain transcriptional information, offers a suitable resource for upcoming analyses of genes or markers related to metabolism and growth that may help improve the preservation of this environmentally relevant species. Further studies can be done analyzing tissue from thoracic and abdominal ganglia within the central nervous system to have a more inclusive and complete set of neural genomic information.

Changes in the behavior of this prawn species may also serve as bioindicators of the presence of contaminants such as plastic derivatives or metals found in rivers. In this study, we thus focused on identifying potential neural markers to evaluate changes in nervous system structure and function that may occur as a result of environmental impacts, such as glutamate neurotransmission. Glutamate, generally eliciting excitatory responses in the crustacean nervous system and at the neuromuscular synapse [35], has been studied extensively at the physiological and pharmacological levels. However, information about glutamate receptors distribution and localization within the nervous system using antibodies is limited. Here, we found five transcripts that multiple sequence alignments and antigenicity analysis suggested as potential candidates for crustaceans-specific antibody production. The transcripts belong to metabotropic glutamate receptors (mGluR) 1 and 4, two transcripts for ionotropic glutamate receptor AMPA 1(GluRIA), and one transcript for the ionotropic glutamate receptor NMDA 2B (GRIN2B).

Metabotropic glutamate receptors are G-protein coupled receptors with seven transmembrane domains that activate molecular cascades to ultimately modify other proteins such as ion channels. mGluR1 is located mainly at the postsynaptic cell [36], and it has been described in mammals [37] and other vertebrates such as *Xenopus laevis* [38], to couple to phospholipase C (PLC) and stimulate the production of diacylglycerol (DAG) and IP3, resulting in the release of intracellular Ca2+ and subsequent cell depolarization and increase in neuronal excitability. mGluR1 is also associated with modulating an increase in NMDA receptor activity [39].

In contrast, mGluR4 is located mainly in the presynaptic cell [36] and it modulates a decrease in NMDA receptor activity, by activating an inhibitory G-protein and preventing the formation of cAMP from ATP [40]. In crustaceans, some of the actions ascribed to these receptors include reducing neurotransmission at neuromuscular junctions [41, 42], inhibiting other glutamatergic synaptic components [22], modulating hormonal signaling pathways [43], reducing anxiety-like behaviors [44], and modulating rhythm in the gastric circuit of the stomatogastric ganglion [45, 46]. At present, there are no studies showing the localization of mGluRs in crustacean neural tissue through immunohistochemistry. We are using the

antibodies designed through analysis of the *M. carcinus* transcriptome described here to address this issue. This approach will help us elucidate the distribution pattern of mGluRs in the prawn´s nervous system and target possible modulatory mechanisms involved in responses of *Macrobrachium* spp. to environmental changes.

Ionotropic glutamate receptors (iGluR), on the other hand, are ligand-gated ion channels that are activated by the neurotransmitter glutamate and are the most prevalent in crustacean CNS neurotransmission [47]. Several studies give these receptors a role in chemoreception of olfactory receptor neurons in the lobster *Panulirus argus* [48], inducible defenses and phenotypic plasticity in the branchiopod *Daphnia pulex* [49] and host recognition in the salmon lice *Lepeophtheirus salmonis* [50]. Three good candidate sequences of iGluR were identified in the *M. carcinus* brain transcriptome: two for AMPA 1 receptors and one for NMDA 2B. iGluR AMPA receptors, the main transducers of excitatory neuromuscular activity in arthropods [51], are mainly located postsynaptically and their most potent agonist is the amino acid L-quisqualate [52], as demonstrated in the crayfish neuromuscular junction. Antibodies against these AMPA receptors can serve as markers of the structure of the neuromuscular synapse in crustaceans and how it can change as a result of experimental manipulations or of natural exposure to environmental variables. In the case of NMDA receptors in crustaceans, some of the functions where they appear to have a role include memory consolidation [53], male sex differentiation [54], visual adaptation [55], reduction of excitatory neurotransmitter release at the neuromuscular junction [26], and participation in neuromuscular control [27]. Some of these studies showed staining of an NMDA 1R-like receptor in the crustacean central and peripheral nervous system using an antibody raised against the mammalian form of the receptor [26–28, 56]. Here, we are offering a potential candidate for crustacean ionotropic glutamate receptor of type NMDA 2B, which has not been evaluated before in a crustacean spp. This provides the advantage of having an antibody targeting a specific sequence derived from the crustacean genomic information, increasing the reproducibility of labeling and decreasing the chances of experiencing non-specific binding with other crustacean proteins, which had been our greatest difficulties with commercially available glutamate receptor antibodies. Moreover, the results presented here are important because not only are they useful for generating antibodies, but also for generating nucleotide sequences as probes for in situ hybridization experiments. This offers a very efficient method to describe changing expression patterns of receptors of interest due to environmental influences.

With the transcriptome contribution and the availability of the antibodies against crustacean glutamate receptors proposed here, the findings of previous physiological and pharmacological studies can be broadened with information regarding the underlying molecular, structural and functional mechanisms. The existence of this tool of genomic information not previously available for this species, represents a reference point to further design and conduct research targeting: 1) possible mechanisms involved in behavioral responses to environmental factors; 2) selection of genes or molecules relevant for metabolism and maintenance for commercial purposes, and 3) the establishment of *Macrobrachium* crustaceans as animal models for neurotoxicology studies.

## Supporting information

**S1 File. Crooke-Rosado et al—S1. Annotated transcriptome.**
(XLSX)

**S2 File. Crooke-Rosado et al—S2. Glu receptor sequences.**
(XLSX)

## Acknowledgments

We would like to thank Dr. Joshua Rosenthal for his efforts and provision of training on using the Eel Pond mRNAseq Protocol for the transcriptome assembly. Also, we thank Dr. Mark W. Miller and Dr. Jacqueline Flores-Otero for the critical reading of this manuscript.

## Author Contributions

**Conceptualization:** Jonathan L. Crooke-Rosado, Maria A. Sosa.

**Data curation:** Jonathan L. Crooke-Rosado.

**Formal analysis:** Jonathan L. Crooke-Rosado, Sara C. Diaz-Mendez, Yamil E. Claudio-Roman.

**Funding acquisition:** Maria A. Sosa.

**Investigation:** Jonathan L. Crooke-Rosado, Nilsa M. Rivera, Maria A. Sosa.

**Methodology:** Jonathan L. Crooke-Rosado, Maria A. Sosa.

**Project administration:** Maria A. Sosa.

**Resources:** Maria A. Sosa.

**Supervision:** Nilsa M. Rivera, Maria A. Sosa.

**Validation:** Jonathan L. Crooke-Rosado.

**Visualization:** Jonathan L. Crooke-Rosado.

**Writing – original draft:** Jonathan L. Crooke-Rosado.

**Writing – review & editing:** Jonathan L. Crooke-Rosado, Maria A. Sosa.

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
