## [Decision Letter · Decision Letter 0]

4 Feb 2021

PONE-D-21-00863

De novo* assembly of the freshwater prawn *Macrobrachium carcinus* brain transcriptome for identification of potential targets to develop antibodies specific for crustacean neural structure and function markers.***

PLOS ONE

Dear Dr. Sosa,

***Thank you for submitting your manuscript to PLOS ONE. After careful consideration, we feel that it is a nice contribution to the field, but does yet not fully meet PLOS ONE’s publication criteria as it currently stands. Therefore, we invite you to submit a revised version of the manuscript that addresses the points raised during the review process. ***

**Please submit your revised manuscript by Mar 21 2021 11:59PM. If you will need more time than this to complete your revisions, please reply to this message or contact the journal office at plosone@plos.org. **

**Please include the following items when submitting your revised manuscript:**

**A rebuttal letter that responds to each point raised by the academic editor and reviewer(s). You should upload this letter as a separate file labeled 'Response to Reviewers'.****A marked-up copy of your manuscript that highlights changes made to the original version. You should upload this as a separate file labeled 'Revised Manuscript with Track Changes'.****An unmarked version of your revised paper without tracked changes. You should upload this as a separate file labeled 'Manuscript'.**

****

We look forward to receiving your revised manuscript.

Kind regards,

Irene Söderhäll

Academic Editor

**PLOS ONE**

Journal Requirements:

***2. ***Please include captions for your Supporting Information files at the end of your manuscript, and update any in-text citations to match accordingly. Please see our Supporting Information guidelines for more information: http://journals.plos.org/plosone/s/supporting-information

****

Reviewers' comments:

**Reviewer's Responses to Questions**

****Comments to the Author****

1. Is the manuscript technically sound, and do the data support the conclusions?

**The manuscript must describe a technically sound piece of scientific research with data that supports the conclusions. Experiments must have been conducted rigorously, with appropriate controls, replication, and sample sizes. The conclusions must be drawn appropriately based on the data presented. **

**Reviewer #1: Yes**

**Reviewer #2: Yes**

**2. Has the statistical analysis been performed appropriately and rigorously? **

**Reviewer #1: N/A**

**Reviewer #2: Yes**

**3. Have the authors made all data underlying the findings in their manuscript fully available?**

**The PLOS Data policy requires authors to make all data underlying the findings described in their manuscript fully available without restriction, with rare exception (please refer to the Data Availability Statement in the manuscript PDF file). The data should be provided as part of the manuscript or its supporting information, or deposited to a public repository. For example, in addition to summary statistics, the data points behind means, medians and variance measures should be available. If there are restrictions on publicly sharing data—e.g. participant privacy or use of data from a third party—those must be specified.**

**Reviewer #1: Yes**

**Reviewer #2: No**

**4. Is the manuscript presented in an intelligible fashion and written in standard English?**

**PLOS ONE does not copyedit accepted manuscripts, so the language in submitted articles must be clear, correct, and unambiguous. Any typographical or grammatical errors should be corrected at revision, so please note any specific errors here.**

**Reviewer #1: Yes**

**Reviewer #2: Yes**

**5. Review Comments to the Author**

**Please use the space provided to explain your answers to the questions above. You may also include additional comments for the author, including concerns about dual publication, research ethics, or publication ethics. (Please upload your review as an attachment if it exceeds 20,000 characters)**

**Reviewer #1: Review of PONE-D-21-00863**

De novo assembly of the freshwater prawn Machrobrachium carcinus brain transcriptome for identification of potential targets to develop antibodies specific for crustacean neural structure and function markers

Authors: Crooke-Rosado et al.

Corresponding author: Maria A. Sosa

Summary. This manuscript is a very important but straight-forward contribution describing the generation of a transcriptome from the brain of a prawn, and the detection of five glutamate receptor transcripts with homologies to known arthropod and mammalian sequences. The work described makes a significant contribution to our knowledge about glutamate receptors crustacean species, and alludes to the production of antibodies and localization efforts that are now underway based on the transcriptome analysis presented. I enjoyed reading this paper very much!

The presentation of this transcriptome work is clear and accessible. The motivations for pursuing this work (use of crustacean species as bioindicators, lack of transcriptomic information about the nervous system and particularly glutamate receptors) are clearly articulated in the introduction. The seven figures are thoughtfully composed to illustrate the results of their analysis, and Figures 3-7 present the glutamine receptor contig sequence data, antigenic propensity values and 3D reconstruction models. The text is exceptionally well organized, and is written using smooth and direct language. I have no major criticisms of this manuscript, but do have a few suggestions for improvement, enumerated below.

1. The title seems long and cumbersome. I suggest perhaps “De novo assembly of the freshwater prawn Macrobrachium carcinus brain transcriptome for identification of potential targets for antibody development”. I think the rest of the original title is actually implicit ---if antibodies are being generated against M. carcinus sequences from brain, these antibodies will by definition be useful as markers in the nervous system.

2. Although glutamate is often referred to as an “excitatory neurotransmitter”, this is actually a misnomer because the effect of a transmitter is defined by the receptor ---not the transmitter. As one example provided in this manuscript, the GluR4 receptor has some inhibitory effects. I suggest rewording “excitatory neurotransmitter” (found on lines 101 and 394), e.g., generally involved in excitatory influences.

3. I believe that the models in part D of Figures 3-7 actually describe the tertiary structure of these contig sequences (not secondary structure as stated in the text and legends), because folds indicative of the side chain interactions are indicated.

Minor editorial suggestions:

Line 54: constituents “of” (not to)

Line 86: American lobster (“American” should be capitalized)

Line 92: “…and pigment-dispersing hormone” (“the” is awkward here)

Lines 105-108: This sentence is long and difficult to read. How about “Feinstein and colleagues….presynaptic membranes of neuromuscular junctions in the crayfish Procambarus clarkii, using electron microscopy and immunocytochemistry with both monoclonal and polyclonal antibodies against the mammalian NMDA receptor.”

Line 110: “specific in the thoracic appendages” (not “at”)

Line 131: “continuously filtered”

Line144: “filter cartridge” does not need to be capitalized

Line 165: “related to” (rather than “with”)

Line 187: “…markers in the prawn CNS and neuromuscular junction…”

Line 211: “a total…was annotated”

Line 220: “associated with” (not “to”)

Line 239: “which contigs” (not “what”)

Line 251: “when examined individually” (not “looked”)

Line 252: “above 1, are predicted to elicit”

Line 265: I suggest that after the comment “showing 6 determinants”, you might include in parenthesis “(peaks in graph)” for those who are not familiar with these plots.

Line 270: “shows reliable sequence conservation” (remove “a”)

Line 341: “European” and “Indian” should be capitalized.

Line 379: use present tense here? “…this is the first study that reports genomic information”

Line 392: “on identifying” (rather than “in”)

Line 397: “…about glutamate receptor distribution and localization…” (possessive not necessary)

Line 406: “Xenopus laevis” ??

Line 430: “…and their most potent agonist…” (should be plural, “their”)

Line 438: This needs clarification, as it’s not clear what a “mammalian-raised” antibody is. I believe you are trying to say “…using an antibody raised against the mammalian form of the receptor…”

Line 440: “This provides the advantage…”

Line 443: “increasing the reproducibility of labeling.”

**Line 444: “With the transcriptome contribution and the availability…”**

**Reviewer #2: This manuscript describes the generation of a transcriptome from the brain of a species of crustacean, the freshwater prawn Macrobrachium carcinus. The authors nicely frame the relevance of their work for their own interests, which is to use sequences of metabotropic and ionotropic glutamate receptors to generate antibodies to examine changes in patterns of expression of these receptors in the brain that are associated with environmental changes and perturbations. I have three major and important comments, all of which the authors should be able to add or clarify in order to round out the paper.**

Major Comments

1. Methods: Need more detail about the tissue used in generating the transcriptome.

a. Was only one transcriptome generated? (Why not more than one, for statistical purposes?)

b. How many animals were used?

c. Why use males only? (what might be the consequences of this with respect to capturing of the glutamate receptors of interest?)

d. What about conditions of the animals used? (size and weight? molt stage? how long were they in the lab after capture in the wild before being used)?

e. It is stated that “brain ganglia” were collected, but what is meant by this? Supraesophageal ganglion? No subesophageal ganglion? Eyestalks with all ganglion and retina? All nerve roots including circumesophageal connectives?

2. Give more information about the intent of the study, especially about the desired specificity of the antibodies and whether the methods have achieved that.

a. It is stated variously that what is desired is “specific antibodies” or “crustacean specific antibodies.” Does the former mean “species specific antibodies”? If either of these, then the method for deciding what amino acid sequences to use may be inadequate. For example, for the antibody for the mGluR1 in Figure 3, the 15-amino acid sequence used for antibody production is identical or virtually so (14 or 15 of the amino acids are identical) in the other three species compared – one other crustacean and two insects. So at best this might be considered a pancrustacean specific antibody, and possibility even less specific than this if the sequence is compared more broadly. For the antibody for mGluR4 in Figure 4, a comparison is made between the sequence for Macrobrachium and three mammals (rodents and primates), with no comparisons of crustaceans, insects, or any other protostomes. So while the sequence similarity between the 15-amino acid sequence used to generate the antibody in Macrobrachium carcinus is relatively dissimilar to the other three sequences, it is impossible to say anything about species-specificity or crustacean-specificity without comparing the Macrobrachium to other crustaceans and insects. And those comparators are available, such as for the other Macrobrachium species with published genomes and transcriptomes, for two other decapod crustaceans in Northcutt et al. 2016 (as cited by the authors), and other crustaceans with sequences in publically available databases. The same applies to the other three GluRs in Macrobrachium, which are compared to only either 3 insect species (Figs. 5 and 6) or three mammal species (Fig. 7). Given this, then while it is clear that there is optimization to produce antibodies that are effective in identifying the receptors of interest in this species, it is not at all clear if the antibodies will be species specific (compared to other crustacean species) or crustacean specific (compared to other pancrustaceans or beyond). In fact, given that there are numerous antibodies to GluRs publicly available, mostly generated to mammalian GluRs, whether these other antibodies would work on Macrobrachium or what might be the relative specificities is not really addressed. Again, I want to emphasize that the approach of the authors is perfectly reasonable to generate antibodies to Macrobrachium carcinus, which I think is the authors’ main goal. However, the authors also mention an interest in making “specific” or “crustacean specific” antibodies, but their methodology does not really make it possible to predict how specific (e.g. species or crustacean specific) their antibodies will be.

b. The authors might mention that their results could be used not only for generating antibodies, but also for generating nucleotide sequences as probes for in situ hybridization. Given the large number of receptors of interest, in situ hybridization could be another method, possibly even more efficient method, to describe changing patterns of express of receptors of interest due to environmental influences.

3. Data availability: the authors state that their data are available in the manuscript and as supplemental data. But they really need to make their data available in more accessible forms, that is, public databases, as is expected from published work. Besides the usual databases that the authors know about, another database of special relevance to the authors is CrustyBase (https://crustybase.org/). Please make the entire transcriptome available, not just these GluR sequences.

Minor Comments

Line 56: “destroying” is not the correct word

Line 63: This reads “The depth of the studies characteristic specificity….is a current research focus.” This does not make sense. Perhaps delete “The depth of the studies”?

Line 75: “SNPs” not “SNP’s”

Line 86: “American” not “American”

Line 110: should be spelled “amphitrite”

Line 123: “spp.” should not be italicized

Daphnia is called a “microcrustacean.” It is small, but what is most relevant is its phylogeny relative to the other species. The authors should use modern phylogenetic terms – perhaps using Schwentner et al. 2017 Current Biology or some other scheme. Daphnia should be called a cladoceran or branchiopod.

**Line 316: “Indian jumping ant” (capital "I")**

**6. PLOS authors have the option to publish the peer review history of their article (what does this mean?). If published, this will include your full peer review and any attached files.**

****

**Reviewer #1: **Yes: **Barbara S. Beltz**

**Reviewer #2: No**

****

**While revising your submission, please upload your figure files to the Preflight Analysis and Conversion Engine (PACE) digital diagnostic tool, https://pacev2.apexcovantage.com/. PACE helps ensure that figures meet PLOS requirements. To use PACE, you must first register as a user. Registration is free. Then, login and navigate to the UPLOAD tab, where you will find detailed instructions on how to use the tool. If you encounter any issues or have any questions when using PACE, please email PLOS at figures@plos.org. Please note that Supporting Information files do not need this step.**

---

## [Author Response · Author response to Decision Letter 0]

21 Mar 2021

Responses to Reviewer #1: (page and line references in parenthesis refer to the “Manuscript with Track Changes” version)

1. Reviewer: The title seems long and cumbersome. I suggest perhaps “De novo assembly of the freshwater prawn Macrobrachium carcinus brain transcriptome for identification of potential targets for antibody development”. I think the rest of the original title is actually implicit ---if antibodies are being generated against M. carcinus sequences from brain, these antibodies will by definition be useful as markers in the nervous system.

Response: When writing the original title, we wanted to be very descriptive to give the reader the fairest idea of what the content of the manuscript would be. However, we agree that the title suggested by reviewer #1 is shorter and at the same time descriptive and reads more clearly. We have thus changed it to: “De novo assembly of the freshwater prawn Macrobrachium carcinus brain transcriptome for identification of potential targets for antibody development” (page 1, lines 2-3).

2. Reviewer: Although glutamate is often referred to as an “excitatory neurotransmitter”, this is actually a misnomer because the effect of a transmitter is defined by the receptor ---not the transmitter. As one example provided in this manuscript, the GluR4 receptor has some inhibitory effects. I suggest rewording “excitatory neurotransmitter” (found on lines 101 and 394), e.g., generally involved in excitatory influences. 

Response: We changed the phrase “being one of the main excitatory neurotransmitters” with what was suggested by the reviewer “being generally involved in excitatory influences” (page 4, line 100-101). We agree and understand that the excitatory response falls on the receptor rather than the neurotransmitter. The sentence in page 17 lines 407-409 was also modified to make this concept clear, eliminating “as one of the most abundant neurotransmitters” and “and the main neurotransmitter” to include the phrase “generally eliciting excitatory responses”. 

3. Reviewer: I believe that the models in part D of Figures 3-7 actually describe the tertiary structure of these contig sequences (not secondary structure as stated in the text and legends), because folds indicative of the side chain interactions are indicated.

Response: This is very true. After revising the figures and looking at the information provided, the models shown in part D of Figures 3-7 represent the compact 3D structure of the tertiary folding caused by the interactions of the different protein domains. Thus, we have replaced the word “secondary” in the text and figure captions with the word “tertiary” (page 11, line 266; page 12, lines 277 and 293; page 13, lines 303 and 320; page 14, lines 330 and 344; page 15, lines 355, 370, and 380).

4. We accept all the minor editorial suggestions made by reviewer #1: 

a. Line 54: constituents “of” (not to)

We replaced “to” with “of” (page 3, line 54).

b. Line 86: American lobster (“American” should be capitalized)

“american” was capitalized - “American lobster” (page 4, line 86)

c. Line 92: “…and pigment-dispersing hormone” (“the” is awkward here)

“the” was eliminated (page 4, line 92).

d. Lines 105-108: This sentence is long and difficult to read. 

This sentence was reworded as follows: “Feinstein and colleagues (26) showed NMDA antibody staining at the presynaptic membranes of neuromuscular junctions in the crayfish Procambarus clarkii.” (page 5, lines 106-109).

e. Line 110: “specific in the thoracic appendages” (not “at”)

“at” was replaced by “in” (page 5, line 112)

f. Line 131: “continuously filtered”

“continuous” was replaced by “continuously” (page 6, line 136)

g. Line 144: “filter cartridge” does not need to be capitalized

uppercase letters in “Filter Cartridge” were changed to lowercase (page 6, line 150)

h. Line 165: “related to” (rather than “with”)

“related with” was replaced by “related to” (page 7, line 172)

i. Line 187: “…markers in the prawn CNS and neuromuscular junction…”

The sentence was reorganized, “prawn” was placed before CNS (page 8, line 194).

j. Line 211: “a total…was annotated” 

“were” was replaced by “was” (page 9, line 219)

k. Line 220: “associated with” (not “to”)

“with” was used instead of “to” (page 10, line 229)

l. Line 239: “which contigs” (not “what”)

“what” was replaced by “which” (page 10, line 249)

m. Line 251: “when examined individually” (not “looked”)

“looked” was replaced by “examined” (page 11, line 261)

n. Line 252: “above 1, are predicted to elicit”

deleted “going” and inserted “predicted” (page 11, line 262)

o. Line 265: I suggest that after the comment “showing 6 determinants”, you might include in parenthesis “(peaks in graph)” for those who are not familiar with these plots.

The parenthesis was inserted after “showing 6 determinants” (page 11, line 276) and also was inserted in all figure’s caption (page 12, line 302; page 14, line 329 and 354; page 15, line 379)

p. Line 270: “shows reliable sequence conservation” (remove “a”)

“a” was eliminated from the sentence (page 12, line 281)

q. Line 341: “European” and “Indian” should be capitalized.

“European” and “Indian” were capitalized as proper nouns (page 13, line 310 and 327; page 14, line 352)

r. Line 379: use present tense here? “…this is the first study that reports genomic information”

Substituted “will” with “reports” (page 16, line 392)

s. Line 392: “on identifying” (rather than “in”)

“in identifying” was replaced by “on identifying” (page 16, line 405)

t. Line 397: “…about glutamate receptor distribution and localization…” (possessive not necessary)

“about its receptor’s distribution…” was replaced with “about glutamate receptor distribution…” (page 17, line 410)

u. Line 406: “Xenopus laevis”??

added the complete species name “Xenopus laevis” (page 17, line 419) 

v. Line 430: “…and their most potent agonist…” (should be plural, “their”)

“it most potent…” was replaced by “their most potent” to make it plural (page 18, line 443)

w. Line 438: This needs clarification, as it’s not clear what a “mammalian-raised” antibody is. I believe you are trying to say “…using an antibody raised against the mammalian form of the receptor…”

This is indeed what we actually wanted to say. The sentence was modified as suggested. (page 18, lines 451 and 452)

x. Line 440: “This provides the advantage…”

“gives” was replaced by “provides” (page 18, line 454)

y. Line 443: “increasing the reproducibility of labeling.”

added “of labeling” after “…increasing the reproducibility” (page 18, line 456)

z. Line 444: “With the transcriptome contribution and the availability…”

The beginning of the sentences was reworded to: “With the transcriptome contribution and the availability…” (page 19, line 463)

Responses to Reviewer #2: (page and line references in parenthesis refer to the “Manuscript with Track Changes” version)

1. Methods: Need more detail about tissue used in generating the transcriptome.

a. Was only one transcriptome generated? (Why not more than one, for statistical purposes?)

Only one transcriptome was generated, as part of a graduate course, and for use to complement an aim of a graduate student´s dissertation project. Our objective was to obtain transcript sequences, at normal physiological conditions, related to ionotropic and metabotropic glutamate receptors to use them as reference for antibody design. To our understanding, more than one transcriptome would be needed when, for example, assessing differential gene expression between two or more conditions (Li, 2019), where statistical analyses would be required. These types of experiments are being planned for the near future, but we believe it is important to make the present transcriptome available to others now.

Li D. Statistical Methods for RNA Sequencing Data Analysis. In: Husi H, editor. Computational Biology [Internet]. Brisbane (AU): Codon Publications; 2019 Nov 21. Chapter 6. Available from: https://www.ncbi.nlm.nih.gov/books/NBK550334/

 doi:10.15586/computationalbiology.2019.ch6

b. How many animals were used?

A total of four (4) biological samples were used in generating the transcriptome (page 5, line 132).

c. Why use males only? (what might be the consequences of this with respect to capturing of the glutamate receptors of interest?) 

Only males were used because of requirements of the collecting permits issued by the Department of Environmental and Natural Resources, that seek to avoid perturbation of the female´s egg release and eclosion cycles in the rivers of Puerto Rico. While there is evidence of sex differences in glutamate receptor gene expression profiles (upregulation or downregulation) in mice and other species, to our knowledge there have been no reports of glutamate receptor types that are expressed solely in one of the sexes. Less is known about sex differences in glutamate receptor expression in crustaceans. For instance, it has been reported that male sex differentiation in decapod and cladoceran crustaceans is mediated by an activation of ionotropic glutamate receptors, especially NMDA subtypes (Toyota et al., 2021, 2015). Nevertheless, since the focus of the present study is to develop markers of crustacean neuromuscular junction synaptic morphology and structure, glutamate receptors identified through a male prawn transcriptome will be adequate, at least for assessing these parameters in male prawns. We will be able to expand on this knowledge and better study potential sex differences in glutamate receptor expression in future experiments where female M. carcinus prawns become more readily available for research, for example by raising in the lab or obtaining through aquaculture farms.

Toyota, K., Miyakawa, H., Hiruta, C., Sato, T., Katayama, H., Ohira, T., & Iguchi, T. (2021). Sex determination and differentiation in decapod and cladoceran crustaceans: An overview of endocrine regulation. Genes, 12(2), 1–16. https://doi.org/10.3390/genes12020305

Toyota, K., Miyakawa, H., Yamaguchi, K., Shigenobu, S., Ogino, Y., Tatarazako, N., … Iguchi, T. (2015). NMDA receptor activation upstream of methyl farnesoate signaling for short day-induced male offspring production in the water flea, Daphnia pulex. BMC Genomics, 16(1), 186. https://doi.org/10.1186/s12864-015-1392-9

d. What about conditions of the animals used? (size and weight? molt stage? how long were they in the lab after capture in the wild before being used)?

Animal Size (cm) Weight (g)

1 14.0 74.70

2 13.3 59.52

3 11.3 41.31

4 13.1 61.88

Animals were not categorized or selected by molt stages. Once the animals arrived at the laboratory facility, they spent ten (10) days in the conditions stated in the methodology until dissection and total RNA extraction (clarification of this has been added, page 6, line 137).

e. It is stated that “brain ganglia” were collected, but what is meant by this? Supraesophageal ganglion? No subesophageal ganglion? Eyestalks with all ganglion and retina? All nerve roots including circumesophageal connectives?

Only the supraesophageal ganglion of the ventral nerve cord was isolated for total RNA extraction (clarification of this has been added, page 6, line 146)

2. Give more information about the intent of the study, especially about the desired specificity of the antibodies and whether the methods have achieved that.

The objective of this study is to develop a de novo transcriptome from the brain (supraesophageal ganglia) of the freshwater prawn M. carcinus to design specific antibodies against glutamate receptors in the crustacean. Our main goal was to generate antibodies that work well in crustacean tissue, since our trials with commercially available antibodies raised in other species have resulted in high non-specific binding and very low consistency of immunohistochemistry staining profiles in Macrobrachium carcinus prawns, the most abundant local species of prawn. Our lab uses this species to assess the impact of environmental perturbations, such as exposure to emerging contaminants or temperature variations, on nervous system structure and function. It is with this intent of finding antibodies better suited for M. carcinus tissue that we decided to explore sequences obtained from a species-specific transcriptome.

The purpose of performing multiple sequence alignments of the glutamate receptor sequences identified in the prawn transcriptome through comparisons with different species (crustacean, insects, human, monkeys, etc.) was to look for those fragments within the sequences with high degree of conservation, suggesting important functional domains for evaluation as potential strong antigenic targets for antibody recognition. With this we are seeking to improve the specificity of the antibodies for recognizing glutamate receptors and not other proteins in the prawn tissue, as has been the case with mammalian-based commercially available antibodies. We understand this point was not clear in the text of our manuscript, as it seemed to imply that we are seeking species specificity rather than receptor specificity. We have thus reworded the original text for better clarity about these points, as follows:

Page 5, lines 116-118 – added the sentence “Unfortunately, in our hands some of the commercially-available glutamate receptor antibodies used with nervous system tissue of Macrobrachium prawns appear to show non-specific binding.”

page 5, lines 120-121 – reworded “specific antibodies” to “antibodies specific to glutamate receptors”

page 5, line 126 – reworded “…develop crustacean nervous system markers” to “…develop nervous system markers that work well in Macrobrachium species…”

Page 10, line 244 – added “…suggesting likely important functional domains,…”

Page 18, lines 456 to 458 – reworded “…decreasing the chances of experiencing non-specific binding with other progeins” to “…increasing the reproducibility of labeling and decreasing the chances of experiencing non-specific binding with other crustacean proteins, which had been our greatest difficulties with commercially available glutamate receptor antibodies.”

Pages 18-19, lines 458 to 462 – added the suggested two sentences: “Moreover, the results presented here are important because not only are they useful for generating antibodies, but also for generating nucleotide sequences as probes for in situ hybridization experiments. This offers a very efficient method to describe changing expression patterns of receptors of interest due to environmental influences. “ 

3. Data availability: Crustybase does not accept transcriptomes that do not include information on differential expression or at least two experimental conditions for comparison. We have instead submitted and registered our transcriptome files and the glutamate receptors sequences with the National Center for Biotechnology Information (NCBI) GenBank, through their tool “Transcriptome Shotgun Assembly Sequence Database” (TSA). We have already created a BioProject (Accession Number: PRJNA716066) for the submission (ID: SUB9330334). Once the NCBI staff completes the required review and provides us with the accession number of the transcriptome file, we will provide it. The link to the transcriptome, presently set as http://www.ncbi.nlm.nih.gov/bioproject/716066, has been added to the manuscript, in page 9, line 216, in addition to the reference to Supplement File 1 (S1 File). The date for release of the database has been preliminarily set for May 21, 2021, and can be modified as needed, to allow time for coordinating the release with the publication date for the manuscript.

4. We accept all the minor editorial suggestions made by reviewer #2: 

a. Line 56: “destroying” is not the correct word

replaced the word “destroying” with “consuming” (page 3, line 56)

b. Line 63: This reads “The depth of the studies characteristic specificity….is a current research focus.” This does not make sense. Perhaps delete “The depth of the studies”?

deleted “The depth of studies characterizing specific…” and replaced it with “Characterization of…” (page 3, line 63)

c. Line 75: “SNPs” not “SNP’s”

The apostrophe was deleted (page 3, line 75)

d. Line 86: “American” not “american” 

“American” was capitalized (page 4, line 86)

e. Line 110: should be spelled “amphitrite”

“y” was replaced by “i” in amphitrite (page 5, line 112)

f. Line 123: “spp.” should not be italicized

“spp” is not italicized anymore (page 5, line 126)

g. Daphnia is called a “microcrustacean.” It is small, but what is most relevant is its phylogeny relative to the other species. The authors should use modern phylogenetic terms – perhaps using Schwentner et al. 2017 Current Biology or some other scheme. Daphnia should be called a cladoceran or branchiopod.

The word “microcrustacean” was replaced by “branchiopod” (page 11, line 254 and 273; page 18, line 439)

h. Line 316: “Indian jumping ant” (capital "I")

“Indian” is capitalized (page 13, line 311 and 327; page 14, line 352)

---

## [Editor Report · Decision Letter 1]

25 Mar 2021

De novo* assembly of the freshwater prawn *Macrobrachium carcinus* brain transcriptome for identification of potential targets for antibody development.***

***PONE-D-21-00863R1***

**Dear Dr. Sosa,**

**We’re pleased to inform you that your manuscript has been judged scientifically suitable for publication and will be formally accepted for publication once it meets all outstanding technical requirements.**

**Within one week, you’ll receive an e-mail detailing the required amendments. When these have been addressed, you’ll receive a formal acceptance letter and your manuscript will be scheduled for publication.**

**An invoice for payment will follow shortly after the formal acceptance. To ensure an efficient process, please log into Editorial Manager at http://www.editorialmanager.com/pone/, click the 'Update My Information' link at the top of the page, and double check that your user information is up-to-date. If you have any billing related questions, please contact our Author Billing department directly at authorbilling@plos.org.**

**If your institution or institutions have a press office, please notify them about your upcoming paper to help maximize its impact. If they’ll be preparing press materials, please inform our press team as soon as possible -- no later than 48 hours after receiving the formal acceptance. Your manuscript will remain under strict press embargo until 2 pm Eastern Time on the date of publication. For more information, please contact onepress@plos.org.**

**Kind regards,**

**Irene Söderhäll**

Academic Editor

**PLOS ONE**
---

## [Editor Report · Acceptance letter]

1 Apr 2021

PONE-D-21-00863R1 

De novo* assembly of the freshwater prawn *Macrobrachium carcinus* brain transcriptome for identification of potential targets for antibody development. ***

Dear Dr. Sosa:

I'm pleased to inform you that your manuscript has been deemed suitable for publication in PLOS ONE. Congratulations! Your manuscript is now with our production department. 

Kind regards, 

on behalf of

Dr. Irene Söderhäll 

Academic Editor

PLOS ONE